# Health Interventions for the Prevention of Dehydration in Agricultural Workers Exposed to Heat Stress: A Systematic Review

**DOI:** 10.3390/healthcare13111232

**Published:** 2025-05-23

**Authors:** Judith Roca, Montserrat Sanromà-Ortiz, Tania Cemeli, Glòria Tort-Nasarre, Ana Lavedán Santamaría, Anna Espart, Carme Cantos-Puig, Carme Campoy

**Affiliations:** 1Department of Nursing and Physiotherapy, University of Lleida, 25199 Lleida, Spain; judith.roca@udl.cat (J.R.); gloria.tort@udl.cat (G.T.-N.); ana.lavedan@udl.cat (A.L.S.); anna.espart@udl.cat (A.E.); carme.campoy@udl.cat (C.C.); 2Health Education, Nursing, Sustainability and Innovation Research Group (GREISI), 25199 Lleida, Spain; 3Health Sciences Library, University of Lleida, Igualada Campus, 08700 Igualada, Spain; carme.cantos@udl.cat

**Keywords:** agricultural workers, dehydration, heat stress, preventive interventions

## Abstract

**Background**: Heat stress in agricultural work is a significant health risk, especially due to dehydration as a result of exposure to heat, physical exertion, and inadequate hydration practices. This problem becomes even more critical when working outdoors, where extreme conditions can intensify the effects. **Objective:** The main objective of the present study was to determine the existing interventions to prevent or mitigate dehydration among agricultural workers exposed to heat stress during their workday outdoors, in both real and simulated contexts. **Methods:** A systematic review was performed in accordance to the PRISMA (Preferred Reporting Items for Systematic Reviews and Meta-Analyses) guidelines. The search strategy combined MeSH terms and an open search in six scientific databases. Relevant studies were selected and data from the interventions were extracted, following the guidelines from the Joanna Briggs Institute (JBI) for systematic reviews. The studies were assessed with the Mixed Methods Appraisal Tool (MMAT) and the GRADE assessment framework. **Results**: Nine studies were included, which focused on interventions such as education programs, cooling devices, and hydration strategies. The results showed that adequate access to water, rest in the shade, the use of cooling bandanas, and hydration backpacks, were effective in reducing dehydration and heat stress among agricultural workers exposed to high temperatures. **Conclusions**: An integrated approach combining education, hydration, and workplace improvements is essential to prevent dehydration and heat stress among agricultural workers. While body cooling is promising, further research and investments in infrastructure are needed to ensure access to safe drinking water, shaded rest areas, cooling technologies, educational initiatives, and health monitoring systems.

## 1. Introduction

The concept of occupational heat stress usually refers to local workplace heat stress associated with environmental factors [1]. The most vulnerable population groups include workers who work outdoors in subtropical or tropical countries, low- or medium-income individuals, and employees in the production sector, such as agriculture, construction, and manufacturing [2]. Agricultural workers exposed to extreme heat face a heightened risk of heat stress, particularly due to prolonged exposure to temperatures often exceeding 30 °C (86 °F), limited mechanization, and, in some cases, the absence of occupational safety programs [2]. The environmental conditions of heat stress during the workday can lead to diverse health problems in agricultural workers, among which we find dehydration and renal failure, among other conditions. These health conditions have been documented in studies conducted across multiple continents [3,4,5,6,7,8], indicating that this is a global issue. Some contributing factors commonly identified include repeated volume depletion due to heat exposure and physical exertion, as well as inadequate hydration practices among agricultural workers [9]. These conditions can lead to kidney damage as a result of recurrent dehydration, reduced renal blood flow, increased tubular reabsorption demands, and elevated uric acid levels [10]. Therefore, thorough research is required to accurately assess the physiological impacts of these factors [11].

Exposure to extreme heat, a high physical workload, and inadequate hydration are key occupational risk factors contributing to dehydration among agricultural workers, alongside cultural or social aspects [12]. Addressing this issue requires not only the consideration of physical risks, but also psychological, social, and well-being dimensions [13]. Despite its relevance, dehydration remains underdocumented in the agricultural sector [8], even as studies highlight both sustained heat exposure and inconsistent regulatory compliance as aggravating factors [14]. In response, effective prevention and treatment strategies for heat-related dehydration must be context-specific and multidisciplinary [3,13]. OSHA emphasizes three core components in heat stress prevention: water access, rest breaks, and shaded areas [15], with water as the primary hydration fluid due to its fundamental role in physiological balance and disease prevention [16].

Scientific evidence shows that agricultural workers exposed to high temperatures during the workday are at considerable risk of dehydration, a risk that is often underestimated. This underscores the importance of implementing multidisciplinary prevention and treatment strategies tailored to environmental, occupational, and health-related factors. To date, no prior literature review has specifically focused on interventions targeting dehydration or heat stress among agricultural workers during their workday. In addition, it must be underlined that the incidence of heat stress will become an increasingly recurring event due to climate change and global warming, which underscores the importance of strengthening the protection of agricultural workers, in particular temporary migrant workers, an especially vulnerable group within the sector [8]. Addressing heat-related risks is essential for meeting some of the Sustainable Development Goals, such as SDG 3 (Good Health and Well-being) and SDG 8 (Decent Work and Economic Growth), and for ensuring the sustainability of the agricultural sector and the resilience of communities [2]. For this, the main objective of the present study was to determine the existing interventions to prevent or mitigate dehydration in agricultural workers exposed to heat stress during their workday outdoors, in both real and simulated contexts.

## 2. Materials and Methods

A systematic review of the literature was performed. Although reviews can address diverse objectives, most are oriented towards assessing an intervention or a treatment [2]; thus, experimental quantitative or quasi-experimental studies were selected. The process described in the Preferred Reporting Items for Systematic reviews and Meta-Analyses literature search extension (PRISMA) [17] was followed (Appendix A). The protocol was registered at Prospero with code CRD42024613308.

For the development of the study, the eight phases described by Aromataris et al., 2024 [18] were followed: (1) Determining the review question, (2) defining the inclusion and exclusion criteria, (3) finding studies through a search, (4) study selection for their inclusion, (5) evaluation of the study quality, (6) data extraction, (7) analysis and synthesis of the pertinent studies, and (8) presentation and interpretation of the results, potentially including a process for establishing the certainty of the set of evidence (through systems such as the Grading of Recommendations, Assessment, Development and Evaluations—GRADE). In particular, phase 8 is specifically developed in the results section.

### 2.1. PHASE 1: Determining the Review Question

A question was formulated based on the PICO (Population, Intervention, Comparison, Outcomes) method “What dehydration prevention and management measures have been applied to agricultural workers exposed to extreme heat conditions?”, and the FINER [19,20] (Feasible, Interesting, Novel, Ethical and Relevant) criteria (Appendix A).

### 2.2. PHASE 2: Definition of the Inclusion and Exclusion Criteria

The inclusion criteria were: (1) Quantitative studies, both experimental and quasi-experimental (pre and post, post-intervention) and mixed (with a quantitative phase), that describe an intervention; (2) Adult participants in an agricultural work situation, exposed to high temperature conditions outdoors or in an simulated environment; and (3) Articles published until November, 2024, without any limits placed on the start date. The exclusion criteria were: (1) Non-primary studies, literature reviews, editorials, or experts opinions; (2) Studies with animals or plants; (3) Studies that do not specifically address dehydration; (4) Studies in which the workday takes place indoors (farms, factories, etc.); (5) Articles that were not in English, Spanish, French, or Italian, and (6) Interventions based on pharmacological treatments.

### 2.3. PHASE 3: Finding Studies Through a Search

To find the studies, the Medical Subject Headings (MeSH) such as dehydration, water–electrolyte balance, agricultural workers’ diseases, hot temperature, and climate change were detailed, as well as open terms such as intervention, fluid intake, outdoor worker, and heat wave, among others.

Then, a search strategy was articulated into two elements: (1) databases, and (2) creation of search formulas. The databases used were: PubMed, Cinahl, Cuiden, US Department of Agriculture, National Agricultural Library, Scopus, WOS, Dialnet, Recercat, Recolecta, TDX, Dart Europe, and Open Access Theses and Dissertations, and the search formulas can be found in Appendix A. This strategy was led by an expert documentarian, a member of the research team (C.Ca).

### 2.4. PHASE 4: Study Selection for Their Inclusion

The articles obtained in the different databases were exported to Rayyan, through the WEB application. This platform accelerates the selection process and is highly usable [21]. A two-stage process was followed: (1) Selection by title and abstract, and (2) selection by complete text. This process was performed by two researchers (T.C., C.C.), and a third one (J.R.) intervened in case of discrepancies. This phase took place in January 2025.

### 2.5. PHASE 5: Evaluation of the Study Quality

The evaluation of the studies was performed by two researchers (C.C., T.C.) following the Mixed Methods Appraisal Tool (MMAT) [22] evaluation criteria for quantitative studies. This decision was made based on the possibility of the different nature of the quantitative studies included, which could be purely quantitative or mixed with quantitative data. The research team defined scores depending on the answer: Yes (25 points), Partial (10 points), and No (0 points); and four qualitative items of categorization were detailed: (a) High: 100 points, (b) moderate: 85 points, (c) low: 70 points, and (d) very low: less than 70 points. With a maximum score of 100 and a minimum of 0. Thus, the higher the score, the higher the methodological quality.

The GRADE system was also used to classify the evidence into four levels: high, moderate, low, or very low, and considering factors such as the study design, the consistency of the results, and risk of bias (Appendix A). Whether or not the items with answers Yes, Partial, or No met the quality criteria was assessed. One point was awarded for a Yes answer and 0 points for a Partial answer, while 1 point was subtracted for a No answer. Four qualitative items of categorization were also detailed: (a) High: 5 points, (b) moderate: more than 3 points, (c) low: 2 points, and (d) very low: less than 2 points [23].

### 2.6. PHASE 6: Data Extraction

The data were extracted from the following dimensions:Basic: author, year, country.Methodological: general objective, study design (experimental and quasi-experimental), participants and context (age, sex, work-related aspects, environmental and housing conditions), and evaluation intervention (variable assessment method).Substantive: interventions (specific actions of the intervention), results (effect measurements), conclusions (narrative synthesis), GRADE.

This process was conducted by three researchers (T.C., C.C., J.R.) and subsequently reviewed by the entire research team (Appendix A).

### 2.7. PHASE 7: Analysis and Synthesis of the Pertinent Studies

The analysis was performed at four levels: (1) Identification of the studies, (2) main characteristics of the articles included, (3) article quality, and (4) interventions to prevent and mitigate dehydration. To provide recommendations based on published interventions for preventing and mitigating dehydration, a descriptive synthesis of the results was chosen. This approach was selected due to the high heterogeneity observed among the included studies, both in the context of intervention development and in the analyzed variables and applied protocols, which precluded a comparative statistical analysis of effectiveness. The synthesis was organized into three types of interventions (educational and behavioral interventions, cooling and prevention of heat stress interventions, and hydration interventions), as the method of result synthesis. Therefore, the findings are initially presented in tabular form, including the internal statistical results of each study (percentages, group comparison tests, among others) and their main conclusions. Subsequently, a detailed narrative presentation of each type of intervention was developed.

## 3. Results

### 3.1. Identification of the Studies

The search in the selected databases provided a result of 558 articles, of which 284 were duplicates. In the first phase, after reviewing the titles and abstracts, 243 articles were excluded, leaving 31 for their complete reading in the second phase. Ultimately, nine articles were found to meet the inclusion criteria for this analysis. The articles excluded did not meet the established criteria, due to aspects such as the lack of participating agricultural workers, work contexts in farms or other buildings, a lack of an intervention, or their classification as projects or editorials (Figure 1).

### 3.2. Characteristics of the Included Studies

The studies were performed in different countries in two continents (America and Asia): two in El Salvador, one in Guatemala, one in Nicaragua, three in the USA, one in Japan, and one in South Korea (Figure 2). The total number of participants was 2046 agricultural workers. One study [24] did not provide the demographic characteristics of the sample, another [25] did not provide specific information (young adults), and the remaining ones provided the mean age (25.4, 28.6, 38, 42, 30, 30.6, 33–35) [24,26,27,28,29,30,31]. Two studies specified sex [27,28], and were the only ones with a higher number of women (90% and 60%), as compared to others, which showed values of 40% [29], 16% [32] or inland 2% and coastland 23% [26].

Two studies conducted in the USA [29,30] detailed that the workers were immigrant Hispanics. It must be detailed that in one of these studies [29] only 8% of the workers spoke English aside from Spanish. Other authors [31] identified the workers as being native to the area.

The work took place outside in agriculture-related areas (fernery, nursery, field crop landscape, sugar mill, sugarcane harvest), and only two studies [28,33] simulated work situations in a laboratory (red pepper harvest, apple orchard). Only two studies identified the length of the workday, one with a mean of 49 h per week (SD 8.5) [30] and the other with a mean workday of 7:40 h [27].

As for the environmental conditions, the articles specified high temperatures. These indicated mean temperatures of 29.3 °C around 1 p.m. [26], from 29.5 to 32.9 °C (maximum of 31.7 and 36.4 °C) [24,31], and a higher mean temperature of 33.5 °C, ranging from 26.7 °C to 36.4 °C [29].

The housing conditions varied according to the studies. In two studies [24,26], the participants were transported in buses or trucks from their homes to the workplace, while in another [30], the workers lived close to the workplace, had previous experience in agriculture, and knew farmers and workers in the area. One study was mixed [31], with production workers from around the area and displaced sugarcane cutters who lived in dormitories in the mill itself. Lastly, some authors [29] detailed that 70% of the participants lived in shack-type dwellings, in a house (13%), or a trailer (18%).

Lastly, it must be detailed that all the included studies, with respect to their design, included one intervention: six experimental (two randomized and four non-randomized), and three quasi-experimental (two post-intervention and one pre-post intervention). Despite the type of design, this last cross-cutting study was included, as it described a specific intervention. Table 1 shows a summary of the data presented in the study according to the following characteristics: (a) basic, (b) methodological, and (c) substantive. In addition, considering the requirements by Aromataris et al. [18] with respect to Phase 8 (presentation and interpretation of the results), the level of evidence according to the Grading of Recommendations, Assessment, Development and Evaluation (GRADE) [23] was included.

### 3.3. Article Quality

The nine articles were analyzed according to the MMAT instrument, and the results found in Table 2 and Table 3 were included according to the type of design. At the global level, it must be noted that 55.56% (five out of nine) obtained a score of 100% (high quality) and 44.44% (four out of nine) obtained a score of 85% (moderate quality), with none of them obtaining a lower score.

### 3.4. Interventions for the Prevention and Mitigation of Dehydration

To enhance the synthesis of the gathered evidence, we have summarized the main findings from the studies included in Table 4, following the structure according to the GRADE approach. The table includes the number of studies, total sample size, observed effects, and a qualitative confidence assessment—based on the CERQual principles—for each of the three intervention categories: Educational, Cooling/Thermal Stress Prevention, and Hydration.

#### 3.4.1. Educational Interventions

One study [29] provided training to six crew leaders on heat-related illnesses (HRI) and heat safety based on the OSHA guidelines. In addition, these crew leaders were equipped with a mobile application to monitor weather conditions and the heat index, enabling them to issue alerts and implement specific protective measures. The adaptive strategies introduced included adjusting work schedules and tasks, taking frequent breaks, wearing hats and light-colored clothing, increasing water intake, resting in shaded areas, and accessing air-conditioned environments during or after the workday to support recovery from heat exposure. However, the study did not establish a direct link between the training received by the crew leaders and the knowledge or behavior of the workers under their supervision. Along this line, the ADELANTE initiative [24] was developed in a platform to promote safe and productive work practices in the sugarcane industry and other sectors. The importance of comprehensive interventions to prevent chronic kidney disease of non-traditional origin (CKDnt) was highlighted.

Another study [31] evaluated the impact of an educational and behavioral intervention with sugarcane workers, both in production and in felling, based on an improved WERS (water, electrolytes, rest, and shade) program. For this, educational resources were used, such as in-person sessions, information posters, and urine color tables for the self-assessment of the degree of hydration. These tools, translated to Spanish and adapted to individuals with a low level of literacy, facilitated the early detection of dehydration signs and promoted preventive actions for avoiding the progression towards kidney failure.

Likewise, the impact of a two-component intervention program was explored [32]: (1) the implementation of a water, rest, and shade (WRS) program, and (2) the advice from consultants specialized in the sugarcane industry. The results underlined the importance of combining education with improvements in workplace conditions, such as the introduction of new cutting protocols and the use of an optimized machete with a more ergonomic and curved handle, which contributed to the reduction in fatigue and the improvement of the worker’s performance.

In addition, an intervention based on the WRS program was applied [26], which included individual backpacks with a capacity of 3 L, mobile rest areas, and programmed rest periods of 10–15 min every 1.5–2 h, along with an additional 45 min period of rest for lunch, representing 25% of the workday. This strategy led to a 25% increase in the consumption of water and a decrease in the symptoms related with heat stress and dehydration, underlining the effectiveness of educational and behavioral interventions, such as, for example, observing the color of urine as an indicator of dehydration, and the need to drink more water. The participants also indicated, in a qualitative manner, that their mood improved with the intervention. Nevertheless, the research assistants during the field work, and also in a qualitative manner, determined the need for more communication to overcome resistance to the WRS intervention.

#### 3.4.2. Cooling and Prevention of Thermal Stress Interventions

One study [27] evaluated the effectiveness of different cooling devices to prevent agricultural workers from exceeding the threshold of a core body temperature of 38.0 °C. The results suggested that cooling bandanas could be a viable option for reducing the risk of hyperthermia as compared to cooling arm sleeves. The bandanas, fabricated with polyvinyl acetate materials and weighing less than a pound (~500 grs), are activated when they are submerged in water for one minute, and maintain their cooling effect for up to four hours. In addition, they can be placed on the head or neck, providing a light and easy-to-use option. On the contrary, although the cooling arm sleeves are also designed to reduce body temperature, they did not show the same efficacy. They tended to be made with heavier and bulkier materials, which can affect the mobility and comfort of the workers, limiting their adoption in agricultural environments.

Along this line, other studies [25,28] have indicated that the combination of multiple cooling devices, such as vests, scarves, and cooling systems integrated into clothing, can be an effective strategy for maintaining body temperature within safe ranges. In particular, the combination of a vest, scarf, and hat was shown to be the most efficient in maintaining body temperature below 38 °C and to reduce heart rate [25]. However, certain limitations were identified, such as the difficulty in maintaining the clothing items and skin dry, as well as the limited duration of the cooling effect of these systems [25]. In another study [28], clothing with integrated cooling technology was used, which helped mitigate heat stress and improve the comfort of the workers under extreme heat conditions.

#### 3.4.3. Hydration Interventions

A recent study [30] showed a high acceptance of hydration backpacks among agricultural workers, which suggests that this intervention could be an efficient tool for promoting adequate hydration. The results indicated that the backpacks, commonly known as camelbacks, were well-received, with frequent use and an improvement in the intake of water reported by the workers. This continuous hydration strategy was also implemented in previous studies. In one of them [26], 3 L devices were used that were originally designed for the USA Special Forces for desert operations, while another study [32] used 3 L backpacks with accessible refills of up to 40 L. These antimicrobial devices only require hand cleaning and allow for a constant supply of water without restricting the worker’s movements. The results indicated that after the intervention, the group of workers who worked inside increased their intake of water from 5.1 L to 6.3 L, which was correlated with a reduction in dehydration symptoms, such as mouth dryness, and a reduction in urine volume. In addition, the workers reported better availability of fresh and clean water, while the researchers observed a higher clarity of the urine samples.

Another study [24] implemented a hydration strategy based on personal thermoses with 5 L of water per worker, access to refrigerators with fresh water, and the administration of envelopes with an electrolyte solution of 300 mL during the rest periods. Each portion contained 7 g of sugar, 50 mg of sodium chloride, and 20 mg of monopotassium phosphate per 100 mL. Additionally, other measures were optimized to improve adaptation to heat, such as an increase in the number of rest periods and the substitution of black tents for green ones, placed in shady areas. On its part, another study [31] evaluated a hydration program for sugarcane cutters that provided 16 L of water and 2.5 L of electrolyte solution per shift, ensuring an adequate provision of liquids to prevent dehydration and to mitigate the impact of heat stress on agricultural workers.

## 4. Discussion

The present systematic review provides an analysis of the interventions implemented to prevent dehydration or its risk in agricultural workers exposed to extreme heat and the risk of heat stress. Of the 558 articles that were initially considered, only nine were selected, as most of the studies were observational without an intervention, were centered on different populations, or did not address other health problems related to agricultural workers. No studies were selected that were published in Europe or Africa. In general terms, the available evidence presents a notable heterogeneity in the study designs, the populations studied, and the specific interventions evaluated.

The working conditions varied according to the geographical context, as shown by studies from the USA [27,29,30] and other regions. In the USA, the OSHA regulates occupational exposure to heat at the state level, establishing reductions during the workday when the temperatures are above 30 °C, as well as other safety measures [30,31]. The regional temperature variability has a direct impact on occupational health, as evidenced by findings from this study, in line with others [13]. This underlines the vulnerability of agricultural workers in tropical and sub-tropical areas, where heat stress and dehydration are significant risks. In addition, the relationship between global warming and the increase in occupational heat stress has been documented, highlighting the need for adapted regulations [13].

The evidence [24,29,31,32] suggests that the educational interventions on heat-related risks and safe practices significantly improve knowledge and work behaviors. Likewise, the importance of continuous training on high outdoor temperature safety is highlighted, especially for less experienced workers, in agreement with other studies [35,36]. Education and awareness programs, along with specific initiatives such as WERS and WER, have demonstrated, through the monitoring of diverse biomarkers (serum creatinine, urine osmolality, serum albumin, eGFR, WBST, and other measurements of temperature) to reduce the impact of heat in vulnerable populations, mitigating the adverse effects on health, especially dehydration and kidney problems [24,31,32]. The high prevalence of kidney problems and dehydration is a recurring finding in the literature [3,4,5,6,7,8]. However, the sustainability of these interventions and their ability to generate lasting changes require more research. For these strategies to be effective, it is fundamental for individuals to perceive their benefits, which can be promoted through public regulations and policies [37]. In addition, promoting health literacy must consider educational and cultural factors, especially in migrant populations with a low level of education [38].

On the one hand, the need to establish CKDu screening protocols and the monitoring of biomarkers is underlined, aside from the evaluation of other health conditions in agricultural workers [24]. On the other hand, monitoring biomarkers and health controls may be challenging due to cultural and social barriers, and difficult access to the health system. A study conducted in 2021 [39] on migrant agricultural workers pointed out that the lack of support staff or independent translators forces workers to depend on their employers to communicate with the doctors, which compromises their privacy and quality of care. This can lead to delays in the treatments, workplace retaliation, and a growing lack of trust in the health system. In addition, many workers perceive that health professionals are not aware of their precarious work conditions, which aggravates their vulnerability and makes access to adequate care more difficult.

The literature reviewed [25,27,28] shows the importance of body cooling to mitigate heat stress in agricultural workers, although there are discrepancies on the efficiency of different devices and their physiological mechanisms. The combination of cooling strategies has been demonstrated to be key in maintaining body temperature and reducing the cardiovascular load [25,28]. In particular, cooling bandanas could be promising, although their long-term effectiveness requires more research [27]. However, the implementation of these devices in rural and jungle areas has some limitations, such as the availability of electricity, logistic difficulties for their distribution, and acceptance by the workers due to factors such as the weight or interference in work tasks. In a qualitative study in 2021 [34], agricultural workers in Florida used cooling devices during their workday outdoors. The cooling bandanas were valued positively, while the arm sleeves were uncomfortable and heavy when they melted.

Hydration through oral intake is fundamental for maintaining body temperature, especially in hot climates and during physical activity. Sweating acts as a key cooling mechanism, but if the water lost is not adequately replenished, it can lead to dehydration, which increases the internal body temperature [40]. Thus, the evidence highlights the importance of guaranteeing water as the main liquid and electrolytes as part of comprehensive strategies, which may include rest in shady areas and education on heat-related risks [10,26,27,31,32]. These interventions have been shown to improve hydration and kidney function, and to reduce heat stress.

The optimum intake of water as the main liquid for hydration varies among studies [10,15], with a recommended amount ranging from 0.8 L/h to 250–300 mL/h, accompanied by rest in shaded areas. In addition, a warning is given about the consumption of drinks with a high content of fructose, due to its relationship with a risk of acute kidney injury [41].

Hydration systems, such as hydration backpacks, have been shown to be effective for maintaining adequate levels of hydration, with a mean consumption of 4.8 L/day in agricultural workers [30] and an increase in hydration of 25% [26]. Nevertheless, their weight can affect performance and the user’s comfort [42]. Alternatively, electrolyte-containing pens can prevent hyponatremia and improve hydration [24]. However, more research is needed on exposure to heat and dehydration, and the possible protective effect of the consumption of electrolyte solutions [4,43].

These interventions, such as access to water and accessible hydration systems at the workplace, are not only effective, but are also easy to implement in diverse agricultural contexts. Their simplicity and viability make them key strategies for mitigating the risks associated to heat stress and dehydration for agricultural workers, especially in environments with high temperatures and vulnerable populations. These hydration guidelines must be planned considering accessibility to water and its quality, as well as the influence of social factors and related power structures [14].

To optimize protection against heat, the combination of hydration with acclimation, adequate clothing, and structured pauses is recommended [13,41]. Some studies [26,29,31,32] suggest implementing three 20-min breaks and a longer break of 60 min for eating in shaded areas. It must be detailed that the studies included in the present review did not address the most adequate type of clothing for agricultural workers. Nevertheless, clothing has an influence on thermal regulation when modifying heat exchange. To minimize heat stress, it must be breathable, with a low capacity to isolate and absorb sweat. In addition, the reflective properties of the material are key aspects in sunny environments, and the ventilation between the skin and the item of clothing favors the dissipation of heat [44].

Lastly, it is essential to highlight that heat stress and dehydration are complex problems that require a comprehensive and coordinated approach to prevent them. The combination of educational strategies, hydration, cooling, and improvement in workplace conditions, such as adequate rest, access to potable water, and shaded areas, is fundamental to safeguard the health of agricultural workers. These interventions must be multidisciplinary and include the active participation of the workers, promoting collaboration between agencies, agricultural communities, and local authorities, with the potential for influencing public policies [38]. The interaction between workplace and environmental conditions must be a key factor in the formulation of stricter policies that not only regulate access to preventive measures, but also guarantee safe and healthy work environments to mitigate the risks of heat stress and dehydration. In addition, it is crucial to conduct additional studies to assess the sustainability and impact of these strategies in different agricultural and environmental contexts, guaranteeing their long-term effectiveness.

### Limitations

The limited number of available articles and their heterogeneity made it impossible to perform an effectiveness analysis such as a meta-analysis. The studies themselves suggest conducting additional research to assess their effectiveness. The certainty of the evidence regarding the effectiveness of interventions to prevent dehydration was moderate, primarily due to imprecision in some small studies and inconsistency in the results across different climatic contexts. The geographical and cultural contexts of the studies varied significantly, which may influence the applicability of the findings to other regions. The differences in environmental conditions, cultural practices, and socioeconomic factors should be considered when interpreting the results. The bias associated with the languages selected as an inclusion criterion must also be detailed, as studies from other geographical areas could have been excluded. In fact, studies were identified that were written in Korean and other languages that were not considered in the analysis. This language bias may have excluded relevant studies that could have provided additional insights into the interventions for preventing dehydration and heat stress.

In addition, the lack of studies with a larger sample and more robust designs restricts the ability to generalize the findings. Some studies included small sample sizes and did not provide detailed demographic information about the participants. The lack of comprehensive data on age, sex, and other relevant characteristics may affect the interpretation of the results and the understanding of how different interventions impact various subgroups of agricultural workers. Other limitations of the analysis include factors such as a gender perspective, cultural influences, and the level of education of the participants, which could have had an influence on the results. The educational background and cultural practices of the participants can affect their perception and adoption of the interventions. Likewise, elements related to the medical history, such as existing chronic or acute conditions, must also be considered. Pre-existing health conditions, such as chronic kidney disease or other comorbidities, can influence the outcomes of the interventions and should be accounted for in future research.

## 5. Conclusions

The current evidence supports a multifaceted strategy that integrates educational efforts on hydration with improvements in workplace conditions to prevent dehydration and heat stress among agricultural workers. These interventions are significantly more effective when actively supported and facilitated by employers.

Behavioral strategies—such as scheduled rest breaks and access to shaded areas—require structural changes in the work environment. While worker education remains essential, it is insufficient without the provision of practical tools and resources by employers. The combination of education and tangible workplace modifications, including access to potable water and cooling equipment, is critical for protecting workers’ health.

Equipment-based interventions, such as cooling bandanas and hydration backpacks, have demonstrated particular effectiveness; for example, cooling bandanas were shown to significantly reduce the risk of hyperthermia. Although personal cooling systems are promising, further research is needed to determine the most suitable devices, particularly considering contextual limitations such as infrastructure. Employer engagement in providing resources and ensuring a safe work environment is vital. Ultimately, reducing heat-related risks in agriculture requires coordinated, sustained efforts from all stakeholders.

## Figures and Tables

**Figure 1 healthcare-13-01232-f001:**
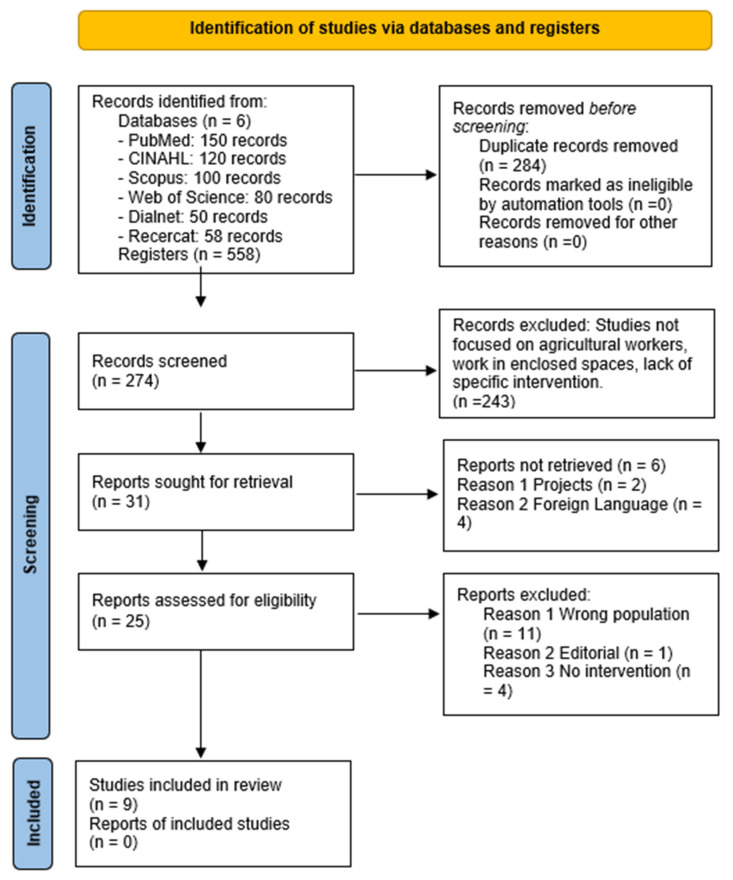
Flowchart of the study selection process according to the PRISMA guidelines.

**Figure 2 healthcare-13-01232-f002:**
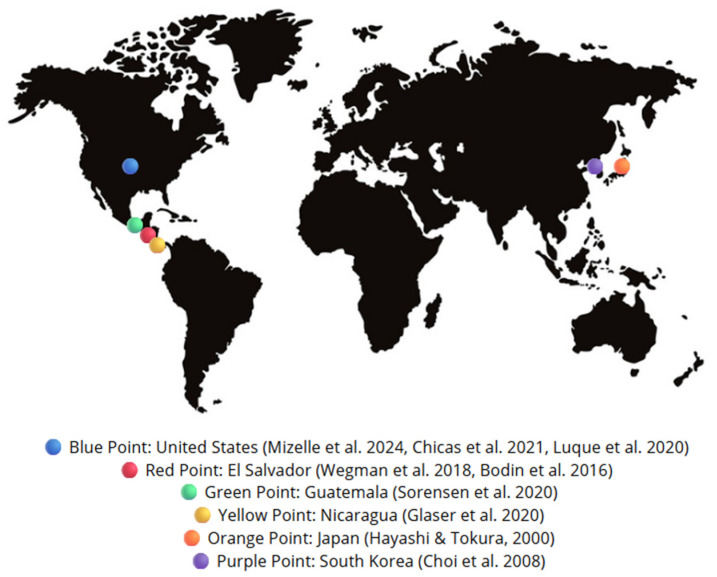
Geographical distribution of study sites [24,25,26,27,28,29,30,31,32].

**Table 1 healthcare-13-01232-t001:** Summary of the characteristics of the studies.

Basic	Methodological				Substantive			
Author, Year, Country	Objective	Design	Participants and Context	Evaluation	Intervention	Results	Conclusions	Grade
Mizelle, 2024 [30], USA	To assess the acceptability of the backpack hydration system intervention for water intake among farmworkers in eastern North Carolina, USA	Post-intervention study	47 male migrant farmworkers from camps in North Carolina	15-question cross-sectional survey on demographics, water intake practices, and system acceptability	Backpack hydration system	Ninety percent of workers considered the backpack acceptable. Fifty-three percent reported using the backpack occasionally, and 28% used it often. They reported an average water intake of 4.8 L/day	Backpack hydration systems are a promising intervention to increase water consumption (frequency and quantity). Further studies are needed to evaluate their effectiveness	H
Chicas R, 2021 [34], USA	To use biomonitoring equipment to examine the effectiveness of selected cooling devices at preventing agricultural workers from exceeding the core body temperature threshold of 38.0 °C (Tc38) and attenuating heat-related illness symptoms	Experimental study	84 Florida farmworkers, during the months of April and May 2018 and 2019	Core body temperature biomonitoring equipment and an accelerometer for physical activity. Pre- and post-work surveys to assess HRI symptoms	Cooling Bandana (Chill-Its^®^ 6700CT Evaporative Cooling Bandana) and Cooling Vest (TechNiche Elite Hybrid Cooling Vest)	The bandana group was less likely to exceed Tc38 (OR = 0.7) compared to the control group. The vest group was more likely to exceed Tc38 (OR = 1.8). The simultaneous use of a vest and bandana showed a similar effect to the control group (OR = 1.3)	Wearing a cooling bandana while working in an agricultural setting has the potential to be protective against exceeding the Tc38 threshold. Future studies with larger sample sizes are needed to determine the effectiveness of cooling interventions	M
Glaser J, 2020 [24], Nicaragua	To assess if the improvement in working conditions related to heat stress was associated with improved kidney health outcomes among sugarcane harvest workers in Chichigalpa, Nicaragua, a region heavily affected by the epidemic of chronic kidney disease of non-traditional origin	Pre-intervention study (harvest 1) and post-intervention study (harvest 2)	Sugarcane harvest workers in Chichigalpa, during the 2017–2018 (525 workers) and 2018–2019 (567) harvest seasons	Serum creatinine measurement before and at the end of harvest. Surveys on demographics, medical history, symptoms, fluid intake, and working conditions.	Improved rest schedules, access to hydration, and shade. Specific recommendations included more breaks in shade tents, improving the taste of water, and distributing electrolyte solutions	In cane cutters, the mean eGFR decline throughout the harvest was 6 mL/min/1.73 m^2^ lower, and the incidence of IKI was 70% lower in the 2018–2019 harvest season compared to the 2017–2018 season. Similar improvements were not observed in the seed cutter and irrigation repair worker groups. Leaders rated the app very positively	The results support the need to improve access to water, rest and shade.	M
Luque J, 2020 [29], USA	(1) To train crew leaders to use the OSHA heat safety tool app and assess their perceptions of the usefulness of the app from the crew leader perspective; and (2) to characterize heat safety knowledge, preventive practices, and perceptions of HRI risk among Hispanic farmworkers	Post-intervention study	101 Hispanic farmworkers and 6 crew leaders in the Florida-Georgia border region during the months of August to October 2018	Surveys on heat safety, HRI knowledge, preventive practices, and risk perceptions	Crew leader training in the use of OSHA’s heat safety application and evaluation of its usefulness	Workers showed little concern about HRI, although 19% had experienced symptoms. Workers with the least awareness were those on H-2A visas (temporary hires), women, and those least concerned about working in hot weather	The need for heat safety training for both crew leaders and farm workers to reduce the risk of HRI, especially among less-experienced workers	H
Sorensen C, 2020 [31], USA and Guatemala	To improve understanding of the natural history of this disease and to evaluate the impact of an educational and behavioral intervention on the trajectories of renal decline among a cohort of Guatemalan sugarcane workers	Experimental and longitudinal study with retrospective and prospective analysis.	517 and 483 sugarcane field workers in Guatemala during the 2016–2017 harvest season, and retrospective data from the 2012–2016 harvest seasons	Pre-employment medical screening data plus serum creatinine. Pre- and post-shift creatinine data. WBGT temperature	WERS program that included additional education, wellness incentives based on hydration status, and medical follow-up	Between 2012 and 2016, the rate of eGFR decline was 0.18 mL/min per 1.73 m^2^ per year for the group with normal kidney function, 2.02 for the group with reduced kidney function, and 7.52 for the group with abnormal kidney function. During the intervention, all groups stabilized or improved their decline trajectory	Early detection of rapid decline in kidney function, combined with appropriate interventions, can halt or slow the progression of kidney failure. Implementing WERS programs and mid-harvest screening protocols for workers at risk for CKDu is necessary	M
Wegman D, 2018 [32], El Salvador	To assess the potential to reduce kidney function damage during the implementation of a water, rest, shade (WRS) and efficiency intervention program among sugarcane workers	Experimental study	117 sugarcane workers in El Salvador in two groups: 60 in the highland group (with intervention) and 57 in the lowland group (without intervention)	Measurement of biomarkers of dehydration and renal function (urinary osmolality, serum albumin, eGFR) at four time points throughout the harvest	A WRS program adapted from OSHA guidelines included water backpacks, mobile shade tents, and scheduled breaks. An efficiency program provided lighter machetes and revised the cutting protocol to reduce lateral movement	Biomarkers showed dehydration and decreased eGFR. The decrease was present in both groups; −10.5 mL/min/1.73 m^2^ (95% CI −11.8–−9.1%), but smaller in the intervention group. During the 5-month harvest, the decrease also occurred in both groups. This decrease appeared to stop after the intervention was introduced	The intervention program appears to reduce the impact of heat stress on both acute and harvest-related kidney function biomarkers	M
Bodin T, 2016 [26], Sweden	(1) To assess the feasibility of providing an intervention modelled on OSHA’s Water, rest, shade program (WRS) during sugarcane cutting and (2) to prevent heat stress and dehydration without decreasing productivity	Phase 1 experimental study with intervention	60 sugarcane workers in El Salvador during the harvest season from November to April	Daily wet bulb globe temperature (WBGT) measurements. Individual production data. Questionnaires and physical examinations. Focus groups	Provision of hydration packs with water, mobile rest areas with shade, and scheduled rest periods	Self-reported water consumption increased by 25%. Symptoms associated with dehydration decreased. Individual daily production increased from 5.1 to 7.3 tons. Focus groups reported a positive perception of the WRS	A WRS intervention is feasible in sugarcane fields and appears to significantly reduce the impact of heat stress conditions on the workforce	M
Choi J, 2008 [25], South Korea and Japan	To evaluate the effects of neck cooling scarves, a cooling vest, a brimmed hat, and the combination of cooling garments, on physiological and subjective responses during the red pepper harvest simulated in a climatic chamber	Experimental study	12 young men simulating the harvest of red peppers in a climate-controlled chamber with a temperature of 33 °C (WBGT)	Measurement of rectal temperature, skin temperature, heart rate, total sweat rate, and subjective responses of participants	Use of different combinations of cooling equipment: neck scarves, cooling vests, and hats with frozen gel. 1. Control (no cooling) 2. Neck scarf A (area: 60 cm^2^) 3. Neck scarf B (area: 154 cm^2^) 4. Hat 5. Vest (area: 606 cm^2^) 6. Hat + Neck scarf B 7. Hat + Vest 8. Hat + Neck scarf B + Vest	The vest, scarf, and hat combination was the most effective in maintaining rectal temperature below 38 °C (*p* < 0.05), reducing heart rate to 120 bpm (*p* < 0.05), and stabilizing skin temperature at 34 °C (*p* < 0.05)	Cooling specific areas of the body, such as the neck and trunk, is effective in reducing heat stress in agricultural jobs exposed to high temperatures	M
Hayashi and Tokura, 2000 [28], Japan	To determine whether our new protective clothing could reduce heat stress on the body, compared with the currently used one	Experimental study	Experiment 1 (E1): 5 young adult women. Experiment 2 (E2): 5 farmers (1 male and 4 females). E1: climate chamber at 28 °C and 60% relative humidity. E2: apple orchard during July, August, and September.	Rectal temperature Heart rate Salivary lactic acid concentration Number of contractions during handgrip exercise Subjective feeling of comfort	Two types of clothing were used: A. Gore-Tex, polyurethane gloves, and rubber boots without a cooling system, B. Pesticide-resistant clothing (100% water-repellent cotton), Long sleeves and pants, Gore-Tex gloves, rubber boots for the feet and ankles, and Gore-Tex around the legs. They were cooled with frozen gel strips on their heads and chests	Rectal temperature was more effectively inhibited in type B (E1) -Heart rate tended to be lower in type B (E1, E2) -Salivary lactic acid at the end of the first exercise was higher in type A (E1) -Hand grip was lower in type A (E1) -Sense of comfort improved in type B (E1, E2)	Newly designed protective clothing (Type B) helps reduce heat stress and improve comfort and fatigue during outdoor work in summer	M

Water, rest, shade (WRS); Water, electrolytes, rest, shade (WERS); Heat-related illness (HRI), Occupational Safety and Health Administration (OSHA); Estimated glomerular filtration rate (eGFR), Incident kidney injury (IKI), Wet bulb globe temperature (WBGT); Chronic kidney disease of unknown origin (CKDu; M (Moderate); H (High).

**Table 2 healthcare-13-01232-t002:** Quantitative randomized controlled trials.

Author, Year	Is There a Clear Description of the Randomization (or an Appropriate Sequence Generation)?	Is There a Clear Description of the Allocation Concealment (or Blinding When Applicable)?	Are There Complete Outcome Data (80% or Above)?	Is There Low Withdrawal/Drop-Out (Below 20%)?	MMAT Score
Chicas R, 2021 [34]	YES 	YES 	YES 	YES 	100%
Choi J, 2008 [25]	YES 	PARTIAL 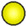	YES 	YES 	85%

**Table 3 healthcare-13-01232-t003:** Quantitative non-randomized controlled, quasi-experimental.

Author, Year	Are Participants (Organizations) Recruited in a Way That Minimizes Selection Bias?	Are Measurements Appropriate (Clear Origin, or Validity Known, or Standard Instrument; and Absence of Contamination Between Groups When Appropriate) Regarding the Exposure/Intervention and Outcomes?	In the Groups Being Compared (Exposed vs. Non-Exposed; with Intervention vs. Without; Cases vs. Controls), Are the Participants Comparable, or Do Researchers Take into Account (Control for) the Difference Between These Groups?	Are There Complete Outcome Data (80% or Above), and, When Applicable, an Acceptable Response rate (60% or Above), or an Acceptable Follow-Up Rate for Cohort Studies (Depending on the Duration of Follow-Up)?	MMAT Score
Mizelle E, 2024 [30]	YES 	YES 	YES 	YES 	100%
Glaser J, 2020 [24]	YES 	YES 	YES 	PARTIAL 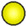	85%
Sorensen C, 2020 [31]	YES 	YES 	YES 	YES 	100%
Luque J, 2020 [29]	YES 	YES 	YES 	YES 	100%
Wegman D, 2018 [32]	YES 	YES 	YES 	PARTIAL 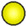	85%
Bodin T, 2016 [26]	YES 	YES 	YES 	PARTIAL 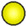	85%
Hayashi and Tokura, 2000 [28]	PARTIAL 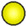	YES 	YES 	YES 	85%

**Table 4 healthcare-13-01232-t004:** Summary of Findings (SoF)—Interventions to prevent dehydration and heat stress among agricultural workers.

Type of Intervention	Number of Studies	Total Sample Size	Observed Effects	Confidence in Evidence (Cerqual)
Educational Interventions	5 Luque et al. (2020) [29], Sorensen et al. (2020) [31], Glaser et al. (2020) [24], Wegman et al. (2018) [32], Bodin et al. (2016) [26]	2.376	Improved awareness of dehydration risks, increased heat illness knowledge, partial behavior changes; some subgroups remained underinformed.	Moderate—Variability in delivery and reliance on self-reported measures.
Cooling and Prevention of Thermal Stress Interventions	3 Chicas et al. (2021) [34], Choi et al. (2008) [25], Hayashi and Tokura (2000) [28]	106	Reduced physiological heat stress, improved comfort and performance, better acceptance of rest practices.	High—Consistent physiological outcomes across contexts.
Hydration Interventions	5 Mizelle et al. (2024) [30], Glaser et al. (2020) [24], Sorensen et al. (2020) [31], Wegman et al. (2018) [32], Bodin et al. (2016) [26],	2.316	Increased fluid access and intake, some improvements in kidney function markers; effectiveness depended on implementation.	Moderate—Some contextual limitations and implementation variability.

## Data Availability

The data analyzed during the current study are not publicly available due to privacy restrictions, but are available from the corresponding authors on reasonable request.

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
