# Peer review of "Health Interventions for the Prevention of Dehydration in Agricultural Workers Exposed to Heat Stress: A Systematic Review"

_healthcare, 2025, doi:10.3390/healthcare13111232_

Round 1
Reviewer 1 Report
Comments and Suggestions for Authors
- The section is overly extensive and, in several passages, diverts attention from the main objective. I recommend shortening it and focusing on the context strictly necessary to understand the purpose of the study.
- Because the authors state they followed the PRISMA guidelines, it is not necessary to include the FINER criteria.
- Moreover, the application of FINER is incorrect in the ethics section: since this is a systematic review without primary data, it should be clarified that ethics committee approval is not required.
- Throughout the manuscript, “systematic review” and “literature review” are used interchangeably inappropriately. It is necessary to standardize on “systematic review” consistently, reserving “literature review” only for explicitly narrative contexts.
- I suggest incorporating the GRADE tool through a Summary of Findings (SoF) table, so that readers can clearly visualize the quality of evidence and the magnitude of effects.
- The flow diagram fails to indicate the databases searched and the number of records retrieved from each. The reasons for exclusion at each stage should also be detailed.
- The title of Figure 1 (“PRISMA”) is imprecise. I propose a more descriptive heading, for example:
- I believe Figure 2 adds no additional value to the manuscript.
- If the authors wish to retain it, they could convert it into a geographic map showing the location of the included studies.
- I have identified discrepancies between the protocol registered in PROSPERO and the methods reported in the manuscript. It is essential to correct and update the registration to accurately reflect the inclusion criteria, variables of interest, and analysis plan.
- The limitations section is too brief. I suggest expanding it.
Author Response
Health interventions for the prevention of dehydration in agricultural workers exposed to heat stress: a systematic review
Manuscript Number: healthcare-3620028
Thank you very much for considering our paper for publication in Healthcare and for taking the time to have it reviewed by experts in the field. We have carefully read and discussed all the reviewers’ comments and made modifications to the manuscript where this was advised. The changes in the text are extensive in the manuscript document. Please find an overview of the changes made below.
We think our manuscript has been improved significantly and we hope you will reconsider it for publication.
This article has been translated and reviewed by a professional translator with expertise in scientific articles.
On behalf of all the (co-)authors.
Reviewer 1:
Comments 1: The section is overly extensive and, in several passages, diverts attention from the main objective. I recommend shortening it and focusing on the context strictly necessary to understand the purpose of the study.
Response 1: Thank you for your comment. We have revised the Introduction section to address the feedback provided by both reviewers. Some ideas have been removed, and certain aspects have been summarized to improve clarity and conciseness. We hope the changes will be viewed positively.
Please do not hesitate to share any further suggestions, as we remain at your disposal.
1.Introduction: Page 1-2, Line 37-80
Comments 2: Because the authors state they followed the PRISMA guidelines, it is not necessary to include the FINER criteria.
Response 2: We thank you for your observation and would like to clarify that the inclusion of the FINER criteria in this systematic review aims to provide a complementary structure that supports the clear and coherent formulation of the research question, as well as other key methodological aspects. While we acknowledge that the PRISMA criteria are essential to ensure transparency and methodological rigor in systematic reviews, we believe that the FINER criteria add value by allowing for the assessment of the feasibility, interest, novelty, ethics, and relevance of the proposed approach.
Therefore, and provided it is not considered inappropriate from an editorial perspective, we would appreciate the opportunity to retain this section as a guiding complement that may be useful for both readers and other researchers interested in the topic.
Comments 3: Moreover, the application of FINER is incorrect in the ethics section: since this is a systematic review without primary data, it should be clarified that ethics committee approval is not required.
Respnse 3: Thank you for your feedback. The text in the ethics section has been revised.
2.Materials and Methods: Page 3, Table 1 (ETHICAL)
Comments 4: Throughout the manuscript, “systematic review” and “literature review” are used interchangeably inappropriately. It is necessary to standardize on “systematic review” consistently, reserving “literature review” only for explicitly narrative contexts.
Response 4: Thank you for your comment. The term systematic review has been used consistently, reserving literature review for narrative approaches.
Comments 5: I suggest incorporating the GRADE tool through a Summary of Findings (SoF) table, so that readers can clearly visualize the quality of evidence and the magnitude of effects.
Response 5: Thank you for your valuable suggestion. In response, we have added a Summary of Findings (SoF) Table 5 in the Results section. This table presents the number of studies per intervention type, total sample size, key reported outcomes, and a confidence assessment adapted from the GRADE-CERQual approach for qualitative evidence. We believe this addition strengthens the clarity, transparency, and practical relevance of our synthesis.
3.Results: Page 12-13, Line 245-50, Table 5
Comments 6: The flow diagram fails to indicate the databases searched and the number of records retrieved from each. The reasons for exclusion at each stage should also be detailed.
Response 6: You are right; Figure 1 has been revised to include the missing information.
Comments 7: The title of Figure 1 (“PRISMA”) is imprecise. I propose a more descriptive heading, for example:
Response 7: Thank you for your suggestion; we have revised the title of Figure 1.
3.Results: Page 6, Line 184
Comments 8: I believe Figure 2 adds no additional value to the manuscript.
Response 8: Thank you for your comment; we will proceed to remove it from the manuscript.
Comments 9: If the authors wish to retain it, they could convert it into a geographic map showing the location of the included studies.
Response 9: We have changed Figure 2, which did not add additional value to the manuscript, and replaced it with this new image you suggested, which we found very interesting.
3.Results: Page 7, Figure 2: Geographical distribution of study sites.
Comments 10: I have identified discrepancies between the protocol registered in PROSPERO and the methods reported in the manuscript. It is essential to correct and update the registration to accurately reflect the inclusion criteria, variables of interest, and analysis plan.
Response 10: Thank you very much for your feedback; we have updated the PROSPERO registration with the latest information.
Comments 11: The limitations section is too brief. I suggest expanding it.
Response 11: Thank you for your input. We have expanded the limitations section as suggested.
4.1 Limitations: Page 17, Line 453-474
We thank the reviewer for their work and time. Without a doubt, it not only helps us to better understand the work, but also significantly improves it.
Reviewer 2 Report
Comments and Suggestions for Authors
Line 32 “better infrastructure are needed” Can you clarify what this is referring to?
Line 40 “it is associated to long periods of extreme temperatures” Can you explain more what this means?
Line 42 “in the workday” Do you mean during?
Line 47 on workers habits of water or alcohol consumption. If you have citations about water consumption studies they would be here. The citation for alcohol use (Morris et al 2024) is not relevant and there is no evidence that workers were studied. This could lead the reader to conclude that ag workers are drinking at work or that their after work activities have a measured effect when they return to the field.
Line 50-52 Also remove the Morris et al 2024 study here.
Line 259 The Luque et al study does not appear to test an intervention. Six Crewleaders were given a phone app (and they evaluated it using app ratings) and all 6 were also provided training in heat safety. I could not find a connection with the worker responses to the survey questionnaire and the training provided to their crewleaders.
Line 465. The conclusion could be strengthened by looking at the success of the interventions that provided material and workplace changes (beyond just education). Cooling bandanas would likely be provided by employers to get wide uptake. All behavioral interventions (adopting the use of a bandana consistently or implementing scheduled breaks and shade) require support and changes to how the agricultural workplace is structured. Education directed at workers does not provide them with the tools needed from their employers and supervisors.
Finally, I am not sure why this often-cited intervention study was not included in this review:
Chavez, Santos, et al. 2022, BMC Public Health. The effect of the participatory heat education and awareness tools (HEAT) intervention on agricultural worker physiological heat strain: results from a parallel, comparison, group randomized study.
Author Response
Health interventions for the prevention of dehydration in agricultural workers exposed to heat stress: a systematic review
Manuscript Number: healthcare-3620028
Thank you very much for considering our paper for publication in Healthcare and for taking the time to have it reviewed by experts in the field. We have carefully read and discussed all the reviewers’ comments and made modifications to the manuscript where this was advised. The changes in the text are extensive in the manuscript document. Please find an overview of the changes made below.
We think our manuscript has been improved significantly and we hope you will reconsider it for publication.
This article has been translated and reviewed by a professional translator with expertise in scientific articles.
On behalf of all the (co-)authors.
Reviewer 2
Comments 1: Line 32 “better infrastructure are needed” Can you clarify what this is referring to?
Response 1: Thank you very much. We have revised the Conclusion (Abstract), adding the clarification regarding infrastructure.
Abstract: Page 1, Line 29-34
Comments 2: Line 40 “it is associated to long periods of extreme temperatures” Can you explain more what this means?
Response 2: We have revised the sentence to include more detailed information.
1.Introduction: Page 1-2, Line 41-44
Comments 3: Line 42 “in the workday” Do you mean during?
Response 3: Thank you for your suggestion. We have included the word "during" to clarify that environmental heat stress conditions affect agricultural workers during their working hours.
1.Introduction: Page 2, Line 44
Comments 4: Line 47 on workers habits of water or alcohol consumption. If you have citations about water consumption studies they would be here. The citation for alcohol use (Morris et al 2024) is not relevant and there is no evidence that workers were studied. This could lead the reader to conclude that ag workers are drinking at work or that their after work activities have a measured effect when they return to the field. Line 50-52 Also remove the Morris et al 2024 study here.
Response 4: Thank you very much for your observation. You are absolutely right regarding the issue of alcohol. We have removed the specific reference and now address the topic in a more general manner.
1.Introduction: Page 2, Line 46-53
Comments 5: Line 259 The Luque et al study does not appear to test an intervention. Six Crewleaders were given a phone app (and they evaluated it using app ratings) and all 6 were also provided training in heat safety. I could not find a connection with the worker responses to the survey questionnaire and the training provided to their crewleaders.
Response 5: You are right—the original wording did not clearly convey the two objectives of the intervention study proposed by Luque (Table 2). Therefore, we have revised the wording to improve clarity.
3.Results: Page 13, Line 256-264
Comments 6: Line 465. The conclusion could be strengthened by looking at the success of the interventions that provided material and workplace changes (beyond just education). Cooling bandanas would likely be provided by employers to get wide uptake. All behavioral interventions (adopting the use of a bandana consistently or implementing scheduled breaks and shade) require support and changes to how the agricultural workplace is structured. Education directed at workers does not provide them with the tools needed from their employers and supervisors.
Response 6: Thank you very much for your comment; your suggestions are absolutely valid. We have revised the entire Conclusion section, incorporating your ideas.
- Conclusions: Page 17-18, Line 476-492
Comments 7: Finally, I am not sure why this often-cited intervention study was not included in this review: Chavez, Santos, et al. 2022, BMC Public Health. The effect of the participatory heat education and awareness tools (HEAT) intervention on agricultural worker physiological heat strain: results from a parallel, comparison, group randomized study.
Response 7: Thank you for your comment and for highlighting the study by Chavez, Santos et al. (2022). We are aware of this work and recognize its relevance in the broader context of heat-related interventions among agricultural workers.
However, this study was not included in our review because it does not specifically address hydration or dehydration—criteria which were part of our predefined inclusion and exclusion criteria. Additionally, as the focus of the intervention was on general heat education rather than hydration-related outcomes, the study did not meet the scope of our database search strategy, which was centered on dehydration and hydration practices.
Thank you once again for your thorough review of our manuscript and for providing valuable suggestions to enhance its quality.
Round 2
Reviewer 1 Report
Comments and Suggestions for Authors
FINER criteria are not applied to a manuscript unless it is part of the review of a research question. If the authors wish to submit it, it should be as supplementary material.
Author Response
Health interventions for the prevention of dehydration in agricultural workers exposed to heat stress: a systematic review
Manuscript Number: healthcare-3620028
REVIEWER 1
Comments 1: FINER criteria are not applied to a manuscript unless it is part of the review of a research question. If the authors wish to submit it, it should be as supplementary material.
Response 1: Thank you for your comment regarding the FINER criteria. We have carefully considered your suggestion and have now moved the FINER criteria analysis to the supplementary material, as you suggested.
Thank you for your continued review of our paper for Healthcare. We have taken your previous comment regarding the FINER criteria into account and have moved this analysis to the supplementary material, as you suggested. This change is reflected in the extensively revised manuscript. We appreciate your guidance and hope the revised version is now suitable for publication.